# Shared decision-making for people living with dementia in extended care settings: a systematic review

Rachel Louise Daly, Frances Bunn, Claire Goodman

## ABSTRACT

**Background**  Shared decision-making is recognised as an important element of person-centred dementia care.

**Objectives**  The aim of this review was to explore how people living with dementia and cognitive impairment can be included in day-to-day decisions about their health and care in extended care settings.

**Design**  A systematic review including primary research relating to shared decision-making, with cognitively impaired adults in (or transferrable to) extended care settings. Databases searched were: CINAHL, PubMed, the Cochrane Library, NICE Evidence, OpenGrey, Autism Data, Google Scholar, Scopus and Medicines Complete (June to October 2016 and updated 2018) for studies published in the last 20 years.

**Results**  Of the 19 included studies 15 involved people with living dementia, seven in extended care settings. People living with cognitive impairment often have the desire and ability to participate in decision-making about their everyday care, although this is regularly underestimated by their staff and family care partners. Shared decision-making has the potential to improve quality of life for both the person living with dementia and those who support them. How resources to support shared decision-making are implemented in extended care settings is less well understood.

**Conclusions**  Evidence suggests that people living with cognitive impairment value opportunities to be involved in everyday decision-making about their care. How these opportunities are created, understood, supported and sustained in extended care settings remains to be determined.

**Trial registration number**  CRD42016035919

## Strengths and limitations of this study

► The review involved a systematic and rigorous search for cross-disciplinary literature relating to shared decision-making for people living with cognitive impairment in extended care settings.

► The majority of studies were conducted in the community rather than in extended care settings.

► Terminology varies across countries, disciplines and professions potentially impacting on the studies retrieved. Including additional search terms around 'choice' may have identified additional relevant papers.

## BACKGROUND

For people living with dementia shared decision-making is increasingly considered crucial in health and care practice[1–4] and is an essential aspect of person-centred care[5]; the fundamental premise of which is that the individual's priorities, interests, abilities and character should inform decision-making.[6] Shared decision-making practice is equally important in extended care. Extended care is defined as residential settings that provide onsite care. This includes supported living, care villages and extra care housing in addition to more traditional care homes with and without nursing.[7]

A dementia diagnosis does not automatically render someone incapable of making a decision[8 9] but for a decision to truly be shared, mutuality must be established.[10] Legislative changes in the last decade have strengthened the rights of people living with dementia to participate in decisions about their care.[11 12] The United Nations Convention on the Rights of People with Disabilities states that 'persons with disabilities enjoy legal capacity on an equal basis with others in all aspects of life.'[13] However, evidence on *how* care decisions are currently made within various legislative frameworks, for and with, people living with dementia, is patchy and often focuses on decisions relating to life events.[11 14] In England and Wales, the Mental Capacity Act (2005) provides a legislative framework to protect and empower people to participate in decisions about their life and care unless there is evidence to the contrary.[15 16] Evidence suggests, however, that staff in extended care often make day-to-day care decisions on behalf of people living with dementia.[17 18] There are situations where people living with dementia in extended care settings decline to participate in decision-making and prefer to relinquish control and delegate to a care

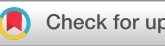

Centre for Research in Primary and Community Care (CRIPACC), University of Hertfordshire, Hatfield, UK

**Correspondence to**
Mrs Rachel Louise Daly;
r.daly2@herts.ac.uk

partner or worker. Reasons identified include: anxiety about ability to participate and reluctance to accept decreasing abilities (reported as self-protection) (see for example, refs [19–21]).

Two reviews have focused on shared decision-making with people living at home with dementia.[21 22] They offer useful insight into degrees of involvement in the decision-making process, what influences that involvement and how people participate.[21 22] In a review of 36 studies of people living with dementia and their carers Miller *et al*[21] identified assorted patterns of decision-making ranging from 'being free to make a choice' through 'supported autonomy' to 'being reliant upon carers'. Larsson's review of 24 studies on community decisions relating to access to care services developed three themes to capture how people living with dementia experience involvement in care decisions: *excluded, prior preferences taken into account* and *current preferences respected*.[22] These reviews usefully bring shared decision-making with people living with dementia into focus but largely excluded people living in extended care settings, who are typically frailer and further along the dementia trajectory. They highlight that decisions for people living with dementia at home invariably involved a care partner/family member. However, family members are often unavailable for many of the day-to-day decisions undertaken in extended care due to time and geographical constraints.

There is increasing recognition that day-to-day decisions are potentially more significant in everyday quality of life than the more noteworthy issues such as treatment decisions or relocation (see for example, refs [23–25]). Building on previous work, this paper considers the evidence on *how* day-to-day decisions are understood and negotiated between people with a cognitive impairment and their (staff and family) carers in extended care settings. The review objectives are presented in box 1.

| Box 1 | Review objectives |
|---|---|

- ► Explore how shared decision-making is understood and/or characterised for people living with dementia and their (staff and family) carers.
- ► Explore the role of (staff and family) carers of people living with dementia in shared decision-making care dyads.
- ► Analyse identified risks and benefits associated with shared decision-making for people with cognitive impairment.
- ► Ascertain empirical evidence for the effectiveness of available shared decision-making resources for people living with dementia.
- ► Seek to understand the barriers and facilitators to effective shared decision-making for people living with dementia and their (staff and family) carers.
- ► Explore the extent to which shared decision-making has been researched in extended care settings.
- ► Identify implications for shared decision-making in dementia care practice, policy and future research.

**Table 1** Study inclusion criteria

| | |
|---|---|
| Publication language | English |
| Publication dates | January 1996 to October 2016 |
| Target population | Adults, aged over 18 years, with any type of cognitive impairment, for example (but not limited to), dementia, learning disabilities, Parkinson's and Huntington's diseases |
| Study setting | Community living at home or extended care settings, for example, supported living, or residential care. Studies must be considered transferable to people living with a cognitive impairment in an extended care setting (eg, the person living with a cognitive impairment must be in receipt of care in addition to that provided by a family care partner and the resource should be suitable for implementation in extended care settings). |
| Study types | All research designs including quantitative and qualitative research |
| Primary outcomes | Any outcome relating to involvement in care planning, satisfaction with decision (eg, care was congruent with expressed choice), quality of life or well-being, and behavioural changes (eg, reduction in distress) |
| Additional outcomes | Family care partner and/or health or care professional satisfaction, and any documented adverse effects (eg, falls, weight loss, adverse outcomes related to medication management) |

## METHODS

A detailed account to the review methods is published elsewhere[26] (PROSPERO registration number CRD42016035919). The review was planned and is reported in accordance with the Preferred Reporting Items for Systematic Reviews and Meta-Analyses guidelines.[27]

Results are presented to reflect relevance to the review question and objectives rather than frequency that the topic arises in the data and must represent a pattern across more than one included study to constitute a theme.[28]

### Study selection and inclusion criteria

Inclusion criteria are presented in table 1. A time limit of 20 years was applied to reflect the rapidly changing literature on involving people living with dementia in their care, published since the seminal works of Tom Kitwood.[29–31] The main focus of the review was on people with dementia but studies relating to adults with other types of cognitive impairment were also included because of the potential for transferable learning.

The term 'care partner' has been adopted to represent a family member or friend who has a caring relationship with the person living with a cognitive impairment. The terms health and/or care professionals, staff or workers are used interchangeably to describe people who are paid to provide health or social care.

**Table 2** Electronic databases searched, and search terms used

| | |
|---|---|
| Electronic databases searched | CINAHL Plus, PubMed, the Cochrane Library, NICE Evidence, OpenGrey, Autism Data, Google Scholar, Scopus and MedicinesComplete |
| MeSH search terms (with subheadings included) | Cognition Disorders searched ((((("dementia"[MeSH Terms] OR "dementia"[All Fields]) OR ("cognitive dysfunction"[MeSH Terms] OR ("cognitive"[All Fields] AND "dysfunction"[All Fields]) OR "cognitive dysfunction"[All Fields] OR ("cognitive"[All Fields] AND "impairment"[All Fields]) OR "cognitive impairment"[All Fields])) OR ("learning disorders"[MeSH Terms] OR ("learning"[All Fields] AND "disorders"[All Fields]) OR "learning disorders"[All Fields] OR ("learning"[All Fields] AND "disability"[All Fields]) OR "learning disability"[All Fields])) OR ("learning disorders"[MeSH Terms] OR ("learning"[All Fields] AND "disorders"[All Fields]) OR "learning disorders"[All Fields] OR ("learning"[All Fields] AND "disorder"[All Fields]) OR "learning disorder"[All Fields])) OR ("autistic disorder"[MeSH Terms] OR ("autistic"[All Fields] AND "disorder"[All Fields]) OR "autistic disorder"[All Fields] OR "autism"[All Fields])) OR ("stroke"[MeSH Terms] OR "stroke"[All Fields])) OR ("brain injuries"[MeSH Terms] OR ("brain"[All Fields] AND "injuries"[All Fields]) OR "brain injuries"[All Fields] OR ("brain"[All Fields] AND "injury"[All Fields]) OR "brain injury"[All Fields])) OR ("neurocognitive disorders"[MeSH Terms] OR ("neurocognitive"[All Fields] AND "disorders"[All Fields]) OR "neurocognitive disorders"[All Fields])) OR ("alzheimer disease"[MeSH Terms] OR ("alzheimer"[All Fields] AND "disease"[All Fields]) OR "alzheimer disease"[All Fields] OR "alzheimer"[All Fields])) AND Shared Decision-making (((((((shared[All Fields] AND ("Decision (Wash D C)"[Journal] OR "decision"[All Fields])) OR ("decision-making"[MeSH Terms] OR ("decision"[All Fields] AND "making"[All Fields]) OR "decision-making"[All Fields])) OR ("community participation"[MeSH Terms] OR ("community"[All Fields] AND "participation"[All Fields]) OR "community participation"[All Fields] OR ("consumer"[All Fields] AND "participation"[All Fields]) OR "consumer participation"[All Fields])) OR ("patient participation"[MeSH Terms] OR ("patient"[All Fields] AND "participation"[All Fields]) OR "patient participation"[All Fields])) OR (("Decision (Wash D C) "[Journal] OR "decision"[All Fields]) AND support[All Fields])) OR (care[All Fields] AND dyad[All Fields])) OR ("patient education handout"[Publication Type] OR "patient education as topic"[MeSH Terms] OR "patient education"[All Fields]))) NOT (((advance[All Fields] AND ("Decision (Wash D C)"[Journal] OR "decision"[All Fields])) OR ("advance directives"[MeSH Terms] OR ("advance"[All Fields] AND "directives"[All Fields]) OR "advance directives"[All Fields] OR ("advance"[All Fields] AND "directive"[All Fields]) OR "advance directive"[All Fields])) OR (advance[All Fields] AND care[All Fields] AND plan[All Fields]))) OR (((((("Dementia"[Mesh] OR "Neurocognitive Disorders"[Mesh] OR "Brain Injuries"[Mesh]) OR "Stroke"[Mesh]) OR "Learning Disorders"[Mesh]) OR "Autistic Disorder"[Mesh]) AND ((((("Decision-making"[Mesh] OR "Decision Support Techniques"[Mesh]) AND "Patient Participation"[Mesh]) OR "Cooperative Behavior"[Mesh]) OR "Physician-Patient Relations"[Mesh]) OR "Patient Education as Topic"[Mesh])) NOT Advance Directives ("Advance Directives"[Mesh] OR "Advance Care Planning"[Mesh]))) NOT Paediatrics ("child'[MeSH Terms] OR "child"[All Fields])) NOT ("Child"[Mesh] OR "Disabled Children"[Mesh]) |
| Alternate free-text search terms | (Cogniti*, Disorder*, Dementia*, Alzheimer*, Neurocogniti* Dis*, Brain Injur*, Autis*, Learning Dis*, Stroke) AND (Shared Decision-making, Deci* Mak*, Patient Participat*, Consumer Participat*, Cooperat*, Decision Support) NOT (Paed* Child*) NOT (Advance Directives, Advance* care planning, Advance* deci*) |
| Google Scholar | The search and screening strategy for Google Scholar was agreed by all three authors. Free-text search terms mirrored other databases. Results were filtered by relevance. The first 20 pages of results, title and abstract were screened (20 results per page). |

## Search strategy

The search strategy drew on a range of cross discipline data sources associated with cognitive impairment and dementia care. Electronic databases searched and an example of the search query for PubMed is given in table 2. Search terms were adapted as appropriate for other databases.

Reference lists of relevant primary and review articles were manually searched to detect additional studies and citation search was performed using the 'cited by' option on Google Scholar, and the 'related articles' option in PubMed. Searches were initially undertaken in June 2016 and updated in October 2016. Lateral searches were completed in November 2016. The search was updated in January 2018.

Electronic search results were downloaded into EndNote. One author (RD) screened references and, where necessary, sought support and independent review from a second author (FB or CG).

## Quality assessment and data extraction

Quality assessment was undertaken using the validated QualSyst framework.[32–34] The QualSyst framework provides comprehensive definitions, instructions and a scoring system for quality scoring of both quantitative and qualitative studies represented as a percentage (with a greater percentage representing higher quality).[34] An ethical approval question was added as this has been highlighted as an essential element of study quality.[35 36]

Quality assessment was carried out by one reviewer (RD), with 10% checked by a second reviewer (FB/CG).

Data were extracted using a structured form that addressed the review objectives. This included information about the study design, participants and outcomes.

## Analysis

Due to heterogeneity, and low numbers, of included quantitative studies, meta-analysis was not considered appropriate therefore results are reported in a narrative format.

Theoretical thematic analysis was undertaken using the research question and review objectives (see box 1) as a framework to map the range of data and identify recurrent themes. This method of synthesising qualitative research draws on work by Braun and Clarke.[28 37] It offers in-depth exploration of the identified themes and areas of interest, which include the roles, resources and people essential to the shared decision-making process relevant to extended care settings.

## RESULTS

Nineteen publications are included[11 12 23 38–53] (see table 3) relating to 18 unique studies (including a doctoral dissertation). Four of the papers are linked as they included a baseline sample drawn from the same pool of participants.[23 41 48 53] However, the studies addressed different questions and are, therefore, included as individual papers. An overview of the

**Table 3** Characteristics of included studies

| Author(s) (year) | Purpose of study | Country | Methods | Quality score (%) | Setting | Participants | Sample size PLWCI | CP | CW | Total | Results or outcomes |
|---|---|---|---|---|---|---|---|---|---|---|---|
| Clarke (2004)[40] | To influence SDM | Australia 3 | Qualitative interviews and observations | 75 | Extended care | PLWD and CW | 13 | | 13 | 26 | Four positive and five negative carer characteristics were identified that impacted on decision-making. |
| Fetherstonhaugh et al (2013)[43] | To understand SDM | | Qualitative interviews | 80 | Home | PLWD | | 6 | | 6 | Three pairs of conflicting attributes identified: (1) subtle support versus taking over; (2) hanging on versus letting go; and (3) being central versus being excluded |
| Miltre et al (2015)[49] | To understand SDM | | Quantitative observed family meetings | 83 | Intermediate care | Older people, CPs and HCPs | 51 | 51 | 2 | 104 | Geriatricians' performance in SDM was mixed; above baseline skill level in some areas and below in others. Longer meetings=better SDM by clinicians. |
| Tyrrell et al (2006)[12] | To measure SDM | France 1 | Qualitative interviews | 83 | Home | PLWD and care partner | 21 | 21 | | 42 | PLWD did not feel listened to and had limited freedom to participate in decision-making. Carers were more satisfied than PLWD. |
| Span (2016)[52] | To facilitate SDM | Holland 1 | Qualitative interviews, focus groups, specialist consultation and workshops | 75 | Home | PLWD, CPs and HCPs | | | | 84 | 18 topics of problems and eight topics addressing decision-making emerged. Only eight topics were identified by both PLWD and care partners. |
| Smebye et al (2012)[11] | To understand SDM | Norway 1 | Mixed methods interviews and observations | 95 | Home and extended care | PLWD, CPs and HCPs | 10 | 10 | 10 | 30 | Care staff do not base mental competence on standardised tests; values and relationships as important as logic. New decision-making categories emerged. Autonomous decision-making occurred but SDM was most typical. |
| Ferm et al (2010)[42] | To facilitate SDM | Sweden 2 | Mixed methods interviews | 95 | Home | PLWD HD | 5 | | | 5 | Talking Mats increased communication but effectiveness depended on conversation topic. |
| Kjellberg (2002)[47] | To understand SDM | | Qualitative interviews | 70 | Home, extended and day care | People living with LD | 23 | | | 23 | Of the nine theoretical combinations of levels of decision-making identified, only five emerged. |
| Bailey et al (2011)[38] | To measure SDM | UK 5 | Quantitative electronic decision-making tasks and questionnaire | 79 | Day services | People living with LD | 24 | | | 24 | Decision-making task performances improved when using the visual aid designed. Although not sustained without the visual aid the improvement was regained when the aid was reintroduced. |
| Boyle (2014)[39] | To measure SDM | | Qualitative creative interaction, observation and interviews | 85 | Home | PLWD and CPs | 5 | 5 | | 10 | Identified that agency related to SDM is demonstrated within six relevant themes. |
| Godwin (2014)[44] | To facilitate SDM | | Mixed methods consultation | 90 | Extended care | PLWD | 34 | | 42 | 76 | Residents were able to demonstrate preferences relating to their environment and enjoyed the consultation process. |
| Murphy and Oliver (2013)[50] | To facilitate SDM | | Mixed methods researcher-facilitated discussion | 65 | Home | PLWD and CPs | 18 | 18 | | 18 | Participants felt more involved in discussions when using Talking Mats although feeling of involvement was significantly higher for carers than for PLWD. |
| Samsi and Manthorpe (2013)[51] | To understand SDM | | Qualitative interviews | 90 | Home | PLWD and CPs | 15 | 15 | | 30 | Three underlying principles identified if decision-making is negotiated and how dynamics changed: importance of autonomy, decision-specific approach and made on someone's behalf described as 'best interest'. |

Continued

**Table 3** Continued

| Author(s) (year) | Purpose of study | Country | Methods | Quality score (%) | Setting | Participants | Sample size | | | | Results or outcomes |
| | | | | | | | PLWCI | CP | CW | Total | |
|---|---|---|---|---|---|---|---|---|---|---|---|
| Feinberg and Whitlatch (2002)[41] | To measure SDM | USA 6 | Quantitative interviews | 80 | Home | PLWD and CPs | 51 | 51 | | 102 | Lower income and carer carer financial strain correlated with how well the PLWD 'felt their carer knew their care wishes (more financial strain=less understanding) |
| Hirschman et al (2005)[45] | To understand SDM | | Interviews | 70 | Home and extended care | PLWD and CPs | 48 | 48 | | 96 | Spousal carer— wife (90%) versus husband (21%). Half care partners of PLWD formally 'lacking capacity' still involved them in decision-making. |
| Horton-Deutsch et al (2007)[46] | To understand SDM | | Mixed methods interviews | 85 | Home | PLWD and CPs | 20 | 20 | | 40 | 75% PLWD had always involved HCP and/or spouse in decisions. 50% PLWD decisions changed 80% in line with CP wishes. Only 55% of dyads congruent throughout. 20% PLWD maintained choice. |
| Menne et al (2008)[48] | To measure SDM | | Quantitative interviews Demographic information including MMSE scores Capacity evaluation | 100 | Home | PLWD and CPs | 217 | 217 | | 434 | PLWD consistently considered themselves to have more involvement in decision-making than their care partners perceived them to be. |
| Menne and Whitlatch (2007)[23] | To measure SDM | | Quantitative secondary data analysis | 86 | Home | PLWD and CPs | 215 | 215 | | 430 | Greater decision-making involvement associated with younger, female, educated, non-spousal CP, fewer months since diagnosis, fewer problems with ADLs, fewer depressive symptoms, and place more importance on autonomy/self-identity. |
| Whitlatch et al (2005)[53] | To measure SDM | | Mixed methods interviews | 100 | Home | PLWD and CPs | 111 | 111 | | 222 | Values and preferences correlated with CP perceptions of PLWD quality of life and involvement in decision-making and with PLWD perception of own quality of life and involvement in decision-making. |

ADL, activities of daily living; CP, care partner; CW, care worker; HCP, healthcare professional; HD, Huntington's disease; LD, learning disability; MMSE, Mini-Mental State Examination; PLWCI, person living with cognitive impairment; PLWD, person living with dementia; SDM, shared decision-making.

screening and selection process is demonstrated in online supplementary figure 1.

## Study quality

Study quality ranged from 65% to 100% (see table 3). No studies were excluded as a result of the quality assessment but a high risk of bias was noted in some studies, such as a measure designed by the same people as the intervention.[42] Eight studies did not provide a clear ethics statement,[12 23 40–43 50 53] including the two studies that scored 100% on QualSyst.[48 53]

## Characteristics of included studies and participants

All included studies (see table 3) were published between 2002 and 2016. The majority (n=14) of studies were published in the last 10 years, suggesting an increasing awareness and interest in shared decision-making for people living with a cognitive impairment that reflects the progression in national and international legislation. All five UK studies were published since the full implementation of the Mental Capacity Act.[15]

Most papers (n=15) focus on people living with dementia[11 12 23 39–41 43–46 48 50–53]; 12 are in 'care dyads' (with a care partner) or 'triads' (with a care partner and a health or care worker). Two papers centre on people with a learning disability,[38 47] one on people living with Huntington's disease,[42] and one includes some people with cognitive impairment.[49] Seven studies represent some participants living in extended care settings,[11 12 40 44 45 47 49 52] while the majority (n=12) relate to people living at home. Of the 19 studies included, only two studies specified *shared decision-making in extended care settings* as their explicit focus.[40 44]

The majority of studies (n=15) employed interviews and/or observations (see table 3). Three of the linked studies used structured interviews and compared the views of people living with a cognitive impairment and their care partners.[41 48 53] One study used structured observations of family meetings.[49] Three studies undertook ethnographic observations of people living with dementia within a care dyad or triad.[11 39 40] Four studies identified person-centred care as a theoretical framework.[39 44 50 52]

A breakdown of whether studies aimed to understand, evaluate interventions or measure shared decision-making is presented in table 3 and discussed within the narrative below. Results are presented in cross-cutting themes that explore decision-making participation or involvement for people living with a cognitive impairment in terms of; how shared decision-making is understood and how participation in decision-making is measured, facilitated and inhibited.

Results are presented to reflect relevance to the review question and objectives rather than frequency that the topic arises in the data.[28]

## Understanding 'shared decision-making' for people living with cognitive impairment

There was no common understanding of what shared decision-making entails and how it can be operationalised for people living with a cognitive impairment. Only one paper offered a definition of shared decision-making. Defining it as an approach that involves patients in making medical decisions with their clinician[54] is influenced by the type and complexity of the decision being made and the perspectives of care partners as well as the person living with cognitive impairment.

Three ways that people living with dementia understand shared decision-making were identified in a phenomenological study[43]: subtle support versus taking over; hanging on versus letting go; and being central versus being marginalised or excluded. One participant described their negative response to being excluded:

> …if someone came in and started telling me how I should run things or do things, I think I would certainly retaliate and not conform to anything they would want to do.[44] (p 147)

A recurrent theme in the qualitative literature reviewed is that for many people living with dementia, it is the participation or 'sharing' in the decision-making process, that is as (if not more) important than making the decision itself.[12 41 43 46] Despite this, Samsi and Manthorpe[51] identified examples of 'substituted decision-making' that essentially excluded the person living with dementia:

> Oh I don't ask her what she wants anymore. I know what she'll say anyway—'anything you like, you decide', so I just do what's best for us both.[52] (p 958)

## Participation in decision-making

The extent to which a person living with dementia is able to participate in decision-making was a focus of many of the included studies. For the purpose of this paper participation in decision-making is subdivided into the degree or level of participation, ability to participate and desire to participate.

### Levels of participation in decision-making

Five studies addressed levels of participation.[11 12 46 47 51] Smebye and colleagues used Thompson's five-point taxonomy of patient involvement and participation in healthcare consultations.[55] Thompson's taxonomy ranges from entirely passive 'non-involvement', through cooperative 'shared decision-making', to independent 'autonomous decision-making'.[55] Smebye *et al* identified two additional elements through care triad observations and interviews: pseudoautonomous (assumptions about a person's decision) and delegated (the person living with dementia expressly delegates their decision-making) when including people living with dementia in care triads.[11] This extended taxonomy is reflected in all five studies exploring the extent of decision-making participation.

Shared decision-making was considered as the most common decision-making pattern by people living with a cognitive impairment. Horton-Deutsch and colleagues[46] explored self-reported participation in decision-making for 20 people living with dementia and their care

partners' using semistructured interviews and a five-point decision-making scale centred on a treatment vignette. The scale ranged from (1) made decisions alone in the past with little input from others and continue to do so, to (5) discussed decisions with spouse in the past and continues to do so. Participants reported that although their self-perceived decision-making changed over time from largely independent historically towards current interdependent (shared) decision-making, the majority (75%) of participants described some level of shared decision-making throughout.[46] Similarly, in a group of 23 people living with varying levels of learning disability, 70% rarely considered themselves entirely independent and relied on shared decision-making in some or all areas of their life.[47]

### Ability to participate

Seven studies[17 38 40 41 45 48 53] explored individuals' ability to participate in decisions often in the context of facilitators or barriers (eg, ref [41]). All the studies presenting measures of decision-making participation compared the responses of the person living with a cognitive impairment with their care partners. They highlighted consistent incongruence between responses, with care partners typically believing the person living with dementia to be less involved than they perceived themselves.[45 53]

Five papers describe and/or evaluate tools that measure aspects of decision-making participation from the perspective of the person living with a cognitive impairment.[12 41 43 44 46] The linked studies (with an overlapping sample ranging from 51 to 217 care dyads) develop, use and evaluate the Decision Control Inventory (DCI) and the Decision-Making Involvement Scale (DMI) to assess and compare everyday care choices of both the person living with a cognitive impairment and their care partner.[23 41 48 53] The DMI Scale measures involvement in everyday decisions such as what food to eat and when to go to bed. It aims to increase participation through communication and improve care planning.[53] The DCI explores the abilities of people living with dementia to control everyday decision-making preferences. The majority of the findings are, however, about home-based care identifying that care partners were chosen as primary substitute decision-makers, but financial strain was correlated with how well people living with dementia felt they were understood.[23 41 48]

Decision-making participation characteristics were explored in a qualitative study of 21 care dyad interviews with older people living with dementia and their care partners.[12] Aspects explored were: information received, being listened to, ability to express an opinion, time allowed for reflection and opportunity to change the decision. People living with dementia felt that they were not given enough time to reflect on decisions and did not feel their views about care provision were listened to. Care partners reported greater satisfaction with the quality of the communication and decision-making process.[12]

Only one study explored the person living with dementia's *desire* to participate in decision-making.[45] Using a vignette, the study examined whether 48 people living with Alzheimer's disease would wish to participate in the decision to take a disease-slowing medication, and what factors (including family) influenced their participation. In total, 92% of people living with dementia wanted to participate in the decision, while only 71% of their care partners thought they would. People living with dementia concentrated on *involvement* in the *process*, while their care partners focused on their relatives' *capacity* to participate. Paradoxically, half of the people living with dementia who were formally assessed as lacking capacity (n=29) had care partners who said their relatives would be involved in decision-making.[45]

### Facilitators and inhibitors of shared decision-making
### Care partners and professionals

Where roles in decision-making are grounded in 'relationships' (or connectedness), the roles of care partners and workers can be facilitators or barriers (see eg, refs [40 43 46 51]).

Only one study explicitly raised the question of who should participate in decision-making and what their roles should be.[52] Yet the impact of the relationship between the person living with dementia and their care partner on decision-making involvement is well documented.[39 45 46 48 51] There is little research focusing on shared decision-making relationships in extended care setting with the majority being undertaken in care dyads within the home environment (see for example, refs [21 22]).

Clarke[40] observed that care workers' characteristics could facilitate or inhibit decision-making involvement for people living with dementia in extended care settings. Positive characteristics included warmth, encouragement with memory and routine. Routine also featured as a negative characteristic if linked to task orientated care. An example is given of a resident who appeared to be asleep and the carer had the person's feet out of bed before explaining what they intended to do[41] (p 20). Other negative characteristics inhibiting shared decision-making were: discouraging independence, depersonalisation and risk adversity. The researchers observed more negative incidents than positive ones and felt that residents' autonomy was compromised.[40] Opportunities for expression of choice were reported to be reduced with the increase of daily care needs in extended care environments.[44 47] Three studies identified reduced shared decision-making associated with: social attitudes, lack of available choices, systems, and care partners and workers identifying more problems than opportunities to involve the person living with dementia.[11 47 52] Optimal decision involvement was achieved by recognising the abilities and rights of the person living with a cognitive impairment as capable of influencing the decision, giving and sharing information, offering support and reinforcing opinions.[11 12 39]

## Tools and resources

Four studies developed and/or evaluated the use of tools and resources to facilitate shared decision-making for someone with cognitive impairment.[38 42 50 52] Two studies evaluated the use of Talking Mats (TM) which are a picture-based, communication framework that allow people to indicate their feelings within a given topic by placing the relevant image on a visual scale. The first study compared TMs with structured and unstructured communication methods with five people in the late stages of Huntington's disease[42] and the second study compared TMs with usual communication methods with 18 people living with dementia and their care partners.[50] Both studies reported improvements in satisfaction with discussions when using TMs, although in one study[50] the feeling of involvement was significantly higher for the carers than for the people living with dementia.

Two studies developed computerised tools to support shared decision-making.[38 52] Span[52] codesigned an interactive web-based tool to promote remote involvement of people living with dementia and their care partners in decision-making around such topics as: social contacts and daily activities, mobility and safety, future care and finances.[52] Bailey and colleagues[38] created a visual aid for people living with a learning disability to facilitate participation in decision-making by presenting different types of information in a uniform format. They argue that this uniformity throughout a decision process could support everyday choices. The participants with a learning disability were trained to use the visual aid and their level of decision-making involvement improved as a result, although ongoing use was required to maintain the improvement.[38]

## Benefits of shared decision-making for people living with a cognitive impairment

Consulting residents living with dementia in an extended care environment, about the care home décor, Godwin[44] noted as an ancillary finding that residents appeared to be 'surprised and pleased' to be asked[45](p 114), arguing that this kind of consultation could enhance their self-esteem and contribute to their quality of life. While this study did not measure the impact of the decision-making process, other studies have identified such benefits as heightened self-esteem, purpose and feeling of self-worth, as outcomes from retaining involvement in decision-making.[43] Ongoing decision-making involvement for people living with dementia is also correlated with reduced depressive symptoms and maintained everyday functioning.[23]

A rerun of the searches revealed three additional potentially relevant papers[56–58]; however, the Alzheimer's Society report[57] did not focus directly on shared decision-making. The qualitative study appraising how people living with dementia make decisions about daycare[56] confirmed the findings of the review in noting the crucial role that professionals can play in facilitating shared decisions. This work appears to be linked with an included

study[52] and would suggest that while there is an ongoing interest in this topic intervention-based work in care home settings remains limited. The study dedicated to shared decision-making in dementia care in care homes[58] is relevant; however, this paper reports only on the care staff perspective of the implementation of the study, and so does not meet the criteria for this review.

## DISCUSSION

The available evidence suggests that people living with cognitive impairment want the opportunity to participate in decision-making about their health and care; it can contribute to a sense of worth and has the potential to improve quality of life.[12 43 45] A lack of opportunity to participate in decision-making is a significant and consistent barrier throughout the literature reviewed (eg, refs [11 40 47 49]). This may be due to confusion about what shared decision-making is in everyday care for people living with cognitive impairment and whether it is in fact opportunities for greater choice that are required. Or it may be professionals' lack of skills in recognising and facilitating people's desire and ability to make a decision.[49] People living with dementia concentrated on *involvement* in the decision-making *process*[12 43 45] but there is a lack of evidence about how the person living with cognitive impairment chooses who they make decisions with and which resources or tools could facilitate people living with a cognitive impairment to *lead* the conversations.

Practical interventions to support and facilitate various aspects of the decision-making process (such as TMs and computer software tools) are reported as having good outcomes for the person living with a cognitive impairment, their care partner and in some cases their health or care professional too.[38 42 50 52] However, current tools predominantly rely on the care partner or professional to identify the decision topic, potentially disempowering the person living with a cognitive impairment. In addition, implementing shared decision-making resources in extended care environments would require care workers being given the time, resource and authority to develop the skills required to use such aids. The included studies fail to provide evidence or discussion of the cost implications associated with embedding shared decision-making for people living with cognitive impairments, or the staff development needed to implement everyday decision-making in extended care settings.

Whether the importance of interdependent relationship between the (family) care partner and the person living with dementia in facilitating or inhibiting shared decision-making at home[39 45 46 48 51] is reflected by the relationship between the person living with dementia and their care staff in extended care settings is not yet understood. However, the frequent underestimation of care partners and workers of the desire and ability of people living with moderate and severe dementia to express preferences about their daily care[11 12 46 47 51] combined with the incongruence in levels of satisfaction in the decision-making

 Daly RL, et al. BMJ Open 2018;8:e018977. doi:10.1136/bmjopen-2017-018977

process between care partners and people living with dementia[12 45 53] raises concerns about the role of consultees under the Mental Capacity Act (2005)[15] in the UK. Although as the process of decision-making may be as important as the decision itself,[12 41 43 44 46] it could be argued that if all parties *perceive* that they have optimum levels of involvement and the desired outcomes are achieved the shared decision-making process is a success.

From the limited evidence available on how relationships in extended care affect shared decision-making it appears that how care staff engage is crucial. They can be enablers or blockers to shared decisions, and this appears to be related to their personality, communication skills and the routines of the workplace.[40 44 49] The implications of the relationships between people living with dementia in extended care and their care staff become increasingly significant as the dementia progresses with care needs increasing and communication capabilities dwindling. A greater understanding of the decision-making needs of the person living with dementia in extended care, and how they can be met, is therefore needed.

### Strengths and limitations
The review involved a systematic and rigorous search for literature relating to shared decision-making for people living with cognitive impairment in extended care settings. As such, this review provides a baseline to inform future research and practice. However, the majority of studies were conducted in the community rather than in extended care settings, quality was variable and there is little evidence on what supports the negotiation of day-to-day decisions between people with a cognitive impairment and their (staff and family) carers in extended care.

The review highlights the difficulties defining what is meant by shared decision-making for people who are cognitively impaired. It is recognised that terminology varies across countries, disciplines and professions potentially impacting on the studies retrieved. Including additional search terms around 'choice' may have identified additional relevant papers that were not identified from terms relating to shared decision-making.

Quality assessment of qualitative studies was changed from the protocol[26] due to access issues so the Qual-Syst tool[34] was used. This tool has limitations particularly related to lack of requirement to assess an ethics statement.

### CONCLUSION
What constitutes shared decision-making in everyday care for people living with cognitive impairment in extended care remains unclear, which in turn leads to confusion about how to embed the process of shared decision-making into everyday practice in extended care. The significance of the interdependent relationships between people living with dementia in extended care and their care staff develops as dementia, care needs and communication

difficulties increase. But whether declining health and function are real or perceived barriers to decision participation remains to be determined, along with the impact everyday shared decision-making would have on the quality of life of people living with dementia in care homes.

People living with cognitive impairment value opportunities to be *involved* in everyday decision-making about their care and involvement in the decision-making process appears to be as important as the decision itself. This desire to share in decision-making is consistently underestimated by care partners and workers, which could have implications for the application of the Mental Capacity Act (2005) in practice. Tools and resources are shown to have a positive impact on decision-making participation; however, in most instances, they do not empower the person living with a cognitive impairment to lead the decision.

Further research is required to understand how opportunities for shared decision-making are created, recognised and understood; and whether they could improve the quality of life for people living with dementia in care homes. Research exploring the relationship between the person living with dementia and their care staff would improve understanding of how shared decision-making can be better facilitated in extended care environments.

### PATIENT AND PUBLIC INVOLVEMENT
The research question and systematic review objectives were presented to the members of the University of Hertfordshire Public Involvement in Research Group (UH PiRG), some of whom have experience of caring for family members with dementia. The group advised on the study design. The results of the review and the resulting study will be presented to the UH PiRG at one of their regular meetings.

**Acknowledgements** The authors are grateful to members of the UH PiRG for their input into the design of this review.

**Contributors** RD wrote the protocol, undertook data extraction, quality assessment and analysis, and wrote the paper. FB and CG supervised and critically appraised all aspects of the process. Important changes from the protocol have been explained. All authors read and approved the final manuscript.

**Funding** This review is undertaken as part of a wider doctoral study focusing on dementia care in care homes which has been funded by the National Institute for Health Research (NIHR) Collaboration for Leadership in Applied Health Research and Care (CLAHRC) East of England. This report presents independent research funded by the NIHR Collaboration for Leadership in Applied Health Research and Care (CLAHRC) East of England, at Cambridgeshire and Peterborough NHS Foundation Trust.

**Disclaimer** The views expressed are those of the author and not necessarily those of the NHS, the NIHR or the Department of Health and Social Care.

**Competing interests** None declared.

**Patient consent** Not required.

**Provenance and peer review** Not commissioned; externally peer reviewed.

**Data sharing statement** No additional data are available.

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
