## [Reviewer comments · BMJ Open]

ARTICLE DETAILS

TITLE (PROVISIONAL)	SHARED DECISION-MAKING FOR PEOPLE LIVING WITH DEMENTIA IN EXTENDED CARE SETTINGS: A SYSTEMATIC REVIEW
AUTHORS	Daly, Rachel; Bunn, Frances; Goodman, Claire

VERSION 1 – REVIEW

REVIEWER	Georg Bollig Palliative Care Team, Medical Department Hospital South Jutland, Denmark
REVIEW RETURNED	26-Sep-2017

GENERAL COMMENTS	A interesting manuscript about an important topic.
--

REVIEWER	Gerard Leavey Ulster University
REVIEW RETURNED	01-Oct-2017

GENERAL COMMENTS	A few caveats. It is somewhat questionable that you have included a range of cognitive impairments in this review. There are considerable differences between a teenager with ID and an elderly person with dementia at the end of life. Decision-making in these contexts are considerably different. I think you might be advised to say something about this. I also think that your strengths and weaknesses are not really thus presented. I also wonder if you considered drawing out what areas of decision-making are relatively problematic. This seems to me to be an important consideration. Lastly, there are some typos that you need to fix (e.g deceasing instead of decreasing) and sentences that are quite banal to the point of meaningless (e.g. everyday decisions are ubiquitous....etc " . otherwise, a useful addition to the literature
---

REVIEWER	Mauro Tettamanti IRCCS-Istituto di Ricerche Farmacologiche Mario Negri, Italy
REVIEW RETURNED	06-Nov-2017

GENERAL COMMENTS	The authors did a systematic review on an important topic, relative to decision making in patients with cognitive impairment/dementia. They did not find many papers on the subject, most of them were works conducted with qualitative instruments, and the aims were rather disparate so they correctly decided to conduct a narrative review. There is however a major point on the reference population I wish to rise.
--

The work is presented with two conflicting titles: "SHARED DECISION-MAKING FOR PEOPLE LIVING WITH DEMENTIA IN EXTENDED CARE SETTINGS: A SYSTEMATIC REVIEW " and "SHARED DECISION-MAKING FOR PEOPLE LIVING WITH COGNITIVE IMPAIRMENT IN EXTENDED CARE SETTINGS: A SYSTEMATIC REVIEW ". This conflict extends to the published protocol: the title of the protocol refers to dementia, but the search strategy mentions autism and other conditions. My preference would be for the strict criterion, since I think different conditions can lead to very different results. Moreover the background is focussed only on dementia and the great majority of the works included in the revision refer to patients with dementia. If the authors want to use also articles on Huntington's disease, cognitive impairment and learning disabilities I think the background should be widened and a limitation should be added pointing to the heterogeneity of the conditions studied.

Minor points

General remarks:

- Correct "Huntingdon's disease" to "Huntington's disease"
- Correct ";" to ":", page 15, lines 37 and 46, page 16, line 31 and in the flow diagram
- Is it possible to summarize the stage of the dementia syndrome? A person with mild dementia has different problems from a person in a severe/very severe stage of dementia, and decision-making can differ a lot.
- "Extended care settings" definition is deferred to page 5, line 49. In my opinion it should appear early in the paper (first lines of the background section).

Abstract

- page 2, line 5 : "Shared decision making ... dementia care." should be under the heading "Background"
- page 2, line 12 (Design) or at the end of the abstract: report systematic review registration number
- page 2, line 21: "Of the 19 included studies 7 involved people living with dementia", while on page 9, line 36 you can read: "Most paper (n=15) focus on people living with dementia": these two sentences seem to contradict each other

Strengths and limitations

- The first two points seem more aims than strengths

Background

- page 4, line 14: I do not understand the sentence "People who are unavailable for many of the day-to-day decisions undertaken in extended care due to time and geographical constraints."

Methods

- Table 1: Primary Outcome Sought: I do not understand the sentence
- page 8, line 44-48: I do not understand the sentence

Results

- page 9, lines 46-48: "Shared decision-making in extended care settings was the focus of only two studies": since the article is on this topic it is not clear why these are the only two studies: do you mean that the other studies focussed on other aims and results on

	shared decision making was reported only as a sub-analysis? - Table 3. The table is highly informative: could you add the reference number (e.g. Clarke, A.M. (2004) [41]) - page 12, lines 21-23: "One participant described their negative response to being excluded": his/her negative response? - In general I found the reported results much interesting, but I expected the results to be written following the 7 points in the review objectives reported in Box 1, while only points 1 and 5 have these titles. Why this discrepancy? PRISMA Checklist: in my opinion items 13, 14, 16, 21 and 23 should have as a result: not applicable
--	--

REVIEWER	Dong Pang Institute for Health Research & School of Healthcare Practice University of Bedfordshire UK
REVIEW RETURNED	12-Jan-2018

GENERAL COMMENTS	It appears that the review has been carried out in a systematic approach in most of review steps, except evidence synthesis. It is unclear how you get your results and conclusions. It may be helpful to add a summary table of included studies, including author's results/conclusions. Regarding quantitative studies, you may present sufficient data for each study in tables, although you don't need to do meta-analysis.
---

VERSION 1 – AUTHOR RESPONSE

Reviewer	Comment	Response
Editorial requirements	Please update the search, as this now over one year old. - Please revise the Strengths and Limitations section (after the abstract) to focus on the methodological strengths and limitations of your study rather than summarizing the results. - Please include the search dates in the abstract.	The search has been updated as per your request. Strengths and limitations have been revised to focus on methods rather than results. Search dates have been included in the abstract as requested.
Reviewer: 1	A interesting manuscript about an important topic.	Thank you.
Reviewer: 2	A few caveats. It is somewhat questionable that you have included a range of cognitive impairments in this review. There are considerable differences between a teenager with ID and an	Thank you for taking the time to thoroughly read and review the manuscript. Your comments are addressed below.

Reviewer	Comment	Response
	elderly person with dementia at the end of life. Decision-making in these contexts are considerably different. I think you might be advised to say something about this. I also think that your strengths and weaknesses are not really thus presented. I also wonder if you considered drawing out what areas of decision-making are relatively problematic. This seems to me to be an important consideration. Lastly, there are some typos that you need to fix (e.g deceasing instead of decreasing) and sentences that are quite banal to the point of meaningless (e.g. everyday decisions are ubiquitous....etc " . otherwise, a useful addition to the literature	The review was undertaken with a view to informing further studies relating to shared decision-making for people living with dementia in care homes. The inclusion of other cognitive impairments was in recognition that there is research from other fields that could provide transferable learning. Identifying specific areas of decision-making that are notably problematic is important but was beyond the scope of this review. The typos and sentences that you highlighted have been revised as per your recommendations.
Reviewer: 3	The authors did a systematic review on an important topic, relative to decision making in patients with cognitive impairment/dementia. They did not find many papers on the subject, most of them were works conducted with qualitative instruments, and the aims were rather disparate so they correctly decided to conduct a narrative review. There is however a major point on the reference population I wish to rise. The work is presented with two conflicting titles: "SHARED DECISION-MAKING FOR PEOPLE LIVING WITH DEMENTIA IN EXTENDED CARE SETTINGS: A SYSTEMATIC REVIEW " and "SHARED DECISION-MAKING FOR PEOPLE LIVING WITH COGNITIVE IMPAIRMENT IN EXTENDED CARE SETTINGS: A SYSTEMATIC REVIEW ". This conflict extends to the published protocol: the title of the protocol refers to dementia, but the search strategy mentions autism and other conditions. My preference would be for the strict criterion, since I think different conditions can lead to very different results. Moreover the background is focussed only on dementia and the great majority of the works included in the revision refer to patients with dementia. If the authors want to use also articles on	Thank you for taking the time to thoroughly read and review the manuscript. Your comments are addressed below: Thank you for highlighting this inconsistency, the title of the review has been revised.

Reviewer	Comment	Response
	Huntington's disease, cognitive impairment and learning disabilities I think the background should be widened and a limitation should be added pointing to the heterogeneity of the conditions studied. Minor points General remarks:  - Correct "Huntingdon's disease" to "Huntington's disease" - Correct ";" to ":", page 15, lines 37 and 46, page 16, line 31 and in the flow diagram - Is it possible to summarize the stage of the dementia syndrome? A person with mild dementia has different problems from a person in a severe/very severe stage of dementia, and decision-making can differ a lot. - "Extended care settings" definition is deferred to page 5, line 49. In my opinion it should appear early in the paper (first lines of the background section). Abstract  - page 2, line 5 : "Shared decision making ... dementia care." should be under the heading "Background" - page 2, line 12 (Design) or at the end of the abstract: report systematic review registration number - page 2, line 21: "Of the 19 included studies 7 involved people living with dementia", while on page 9, line 36 you can read: "Most paper (n=15) focus on people living with dementia": these two sentences seem to contradict each other Strengths and limitations  - The first two points seem more aims than strengths Background  - page 4, line 14: I do not understand the sentence "People who are unavailable for many of the day-to-day decisions undertaken in extended care due to time and geographical constraints." Methods	The review was undertaken with a view to informing further studies relating to shared decision-making for people living with dementia in care homes. The inclusion of other cognitive impairments was in recognition that there was potentially research from other fields that could provide transferable learning. This should have been made clearer. A sentence has been added in the 'Study selection and inclusion criteria' section to explain this as per your recommendation. Spellings and grammar have been corrected. It is not possible to summarize the stage of dementia in a way that is meaningful to the subject as studies measure the stage of dementia and other cognitive impairment in different ways. In addition, MMSE score which has been called into question for decision-making participation and abilities (see e.g. Milte 2015, Pratt and Wilkinson 2001). The definition of extended care has been moved to the end of the 1st paragraph in the background.

Reviewer	Comment	Response
	- Table 1: Primary Outcome Sought: I do not understand the sentence - page 8, line 44-48: I do not understand the sentence Results - page 9, lines 46-48: "Shared decision-making in extended care settings was the focus of only two studies": since the article is on this topic it is not clear why these are the only two studies: do you mean that the other studies focussed on other aims and results on shared decision making was reported only as a sub-analysis? - Table 3. The table is highly informative: could you add the reference number (e.g. Clarke, A.M. (2004) [41]) - page 12, lines 21-23: "One participant described their negative response to being excluded": his/her negative response? - In general I found the reported results much interesting, but I expected the results to be written following the 7 points in the review objectives reported in Box 1, while only points 1 and 5 have these titles. Why this discrepancy? PRISMA Checklist: in my opinion items 13, 14, 16, 21 and 23 should have as a result: not applicable	Abstract Page 2, line 5 : "Shared decision making ... dementia care." moved to a heading "Background" Page 2, line 12 (Design) systematic review registration number added page 2, line 21: and page 9, line 36 Whilst these two sentences appear to contradict each other they were designed to highlight that whilst 15 studies focused on people living with dementia only 7 studies included people living with dementia in extended care. Strengths and limitations have been revised to focus on methods rather than results. This sentence relates to family care partners who are often unavailable in extended care settings. The sentence has been revised to make the meaning clearer. The sentence has been revised. The sentence has been revised. Only 2 studies were identified that had the primary focus of shared decision-making in extended care. The sentence has been revised to

Reviewer	Comment	Response
		make this clearer. The reference numbers have been added as per your suggestion. The response is directly below in italics. The review objectives were identified prior to data extraction and that presenting the results clearly under the objective headings proved impossible. This has now been explained under Characteristics of included studies and participants The PRISMA Checklist: has been revised and represented.
Reviewer: 4	It appears that the review has been carried out in a systematic approach in most of review steps, except evidence synthesis. It is unclear how you get your results and conclusions. It may be helpful to add a summary table of included studies, including author's results/conclusions. Regarding quantitative studies, you may present sufficient data for each study in tables, although you don't need to do meta-analysis.	Thank you for taking the time to read and review the manuscript. The section on analysis has been revised and an extra column has been included in table 3 as per your recommendation.

VERSION 2 – REVIEW

REVIEWER	Gerard Leavey Ulster University
REVIEW RETURNED	02-Mar-2018
GENERAL COMMENTS	This is an important but seldom explored area. This is an excellent paper which addresses the challenges of decision-making in

	cognitive impairment. The only slight disjunct I see is between the study objective which asks the question "how people living with dementia can be included in decision-making" and the results and conclusions which provide a somewhat different answer. Also, the results and conclusions text are too similar.
--	---

VERSION 2 – AUTHOR RESPONSE

Reviewer	Comment	Response
Reviewer: 2	This is an important but seldom explored area. This is an excellent paper which addresses the challenges of decision-making in cognitive impairment. The only slight disjunct I see is between the study objective which asks the question "how people living with dementia can be included in decision-making" and the results and conclusions which provide a somewhat different answer. Also, the results and conclusions text are too similar.	Thank you for taking the time to review the manuscript. Your comments are addressed below. The study objective aimed to identify "how people living with dementia can be included in decision-making?" The results presented identify that there are a number of tools and resources that have been developed to support, measure and facilitate involvement in decision-making. On p10 we explain that "results are presented in cross-cutting themes that explore decision-making participation or involvement for people living with a cognitive impairment in terms of; how shared decision-making is understood and how participation in decision-making is measured, facilitated and inhibited." I hope that helps to clarify how the results answer the question. The conclusions text has been amended.